# Semantic Abstraction: Open-World 3D Scene Understanding from 2D Vision-Language Models

**Huy Ha**        **Shuran Song**

Columbia University

semantic-abstraction.cs.columbia.edu

**Abstract:** We study open-world 3D scene understanding, a family of tasks that require agents to reason about their 3D environment with an open-set vocabulary and out-of-domain visual inputs – a critical skill for robots to operate in the unstructured 3D world. Towards this end, we propose Semantic Abstraction (SemAbs), a framework that equips 2D Vision-Language Models (VLMs) with new 3D spatial capabilities, while maintaining their zero-shot robustness. We achieve this abstraction using relevancy maps extracted from CLIP, and learn 3D spatial and geometric reasoning skills on top of those abstractions in a semantic-agnostic manner. We demonstrate the usefulness of SemAbs on two open-world 3D scene understanding tasks: 1) completing partially observed objects and 2) localizing hidden objects from language descriptions. SemAbs can generalize to novel vocabulary for object attributes and nouns, materials/lighting, classes, and domains (i.e., real-world scans) from training on limited 3D synthetic data.

**Keywords:** 3D scene understanding, out-of-domain generalization, language

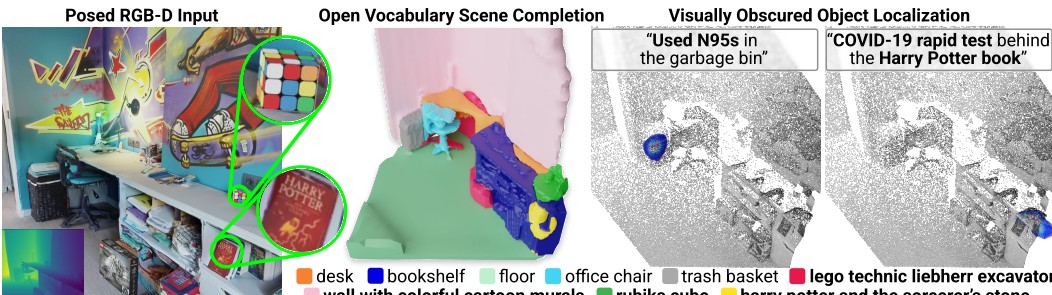

Figure 1: **Open-World 3D Scene Understanding**. Our approach, Semantic Abstraction, unlocks 2D VLM's capabilities to 3D scene understanding. Trained with a small simulated data, our model generalizes to unseen classes in a novel domain (*i.e.*, real world scans), from small objects like "rubiks cube", to long-tail concepts like "harry potter", to hidden objects like the "used N95s in the garbage bin".

## 1 Introduction

To assist people in their daily life, robots must recognize a large, context-dependent, and dynamic set of semantic categories from their visual sensors. However, if their 3D training environment contains only *common* categories like desks, office chairs, and bookshelves, how could they represent and localize *long-tail* objects, such as the "used N95s in the garbage bin" (Fig 1)?

This is an example of an open-world 3D scene understanding task (Fig. 2), a family of 3D vision-language tasks which encompasses open-set classification (evaluation on novel classes) with extra generalization requirements to novel vocabulary (*i.e.*, object attributes, synonyms of object nouns), visual properties (*e.g.* lighting, textures), and domains (*e.g.* sim v.s. real). The core challenge in such tasks is the limited data: all existing 3D datasets [1–6] are limited in diversity and scale compared to their internet-scale 2D counterparts [7–9], so training on them does not prepare robots for the open 3D world.

On the other hand, large-scale 2D vision language models (VLM) [7–11] have demonstrated impressive performance on open-set image classification. Through exposure to many image-caption pairs from the internet, they have acquired an unprecedented level of robustness to visual distributional shifts and a large repertoire of visual-semantic concepts and vocabulary. However, fine-tuning these pretrained models reduces their open-set classification robustness [7, 10, 12, 13]. Compared to internet-scale image-caption data, the finetuning datasets, including the existing 3D datasets [1–6], show a significant reduction in scale

6th Conference on Robot Learning (CoRL 2022), Auckland, New Zealand.

and diversity, which causes finetuned models to become task-specialized and lose their generality. Hence, we investigate the following question:

*How can we equip 2D VLMs with new 3D capabilities,*
*while maintaining their zero-shot robustness?*

We propose **Semantic Abstraction (SemAbs)**, a framework for tackling visual-semantic reasoning in open-world 3D scene understanding tasks using 2D VLMs. We hypothesize that, while open-world visual-semantic reasoning requires exposure to internet-scale datasets, 3D spatial and geometric reasoning is tractable even with a limited synthetic dataset, and could generalize better if learned in a semantic-agnostic manner. For instance, instead of learning the concept of "behind the Harry Potter book", the 3D localization model only needs to learn the concept of "behind *that object*".

To achieve this abstraction, we leverage the relevancy maps extracted from 2D VLMs and used them as the "abstracted object" representation that is agnostic to their semantic labels. The SemAbs module factorizes into two submodules: 1) A semantic-aware wrapper that takes an input RGB-D image and object category label and outputs the relevancy map of a pre-trained 2D VLM model (i.e., CLIP [7]) for that label, and 2) a semantic-abstracted 3D module  that uses the relevancy map to predict 3D occupancy. This 3D occupancy can either represent the 3D geometry of a partially observed object or the possible 3D locations of a hidden object (*e.g.* mask in the trash) conditioned on the language input.

While we only train the 3D network on a limited synthetic 3D dataset, it generalizes to any novel semantic labels, vocabulary, visual properties, and domains that the 2D VLM can generalize to. As a result, the SemAbs module inherits the VLM's visual robustness and open-set classification abilities while adding the spatial reasoning abilities it lacks [14–16]. In summary, our contributions are three-fold[1]:

- **Semantic Abstraction**, a framework for augmenting 2D VLMs with 3D reasoning capabilities for open-world 3D scene understanding tasks. By abstracting semantics away using relevancy maps, our SemAbs module generalizes to novel semantic labels, vocabulary, visual properties, and domains (*i.e.*, sim2real) despite being trained on only a limited synthetic 3D dataset.
- **Efficient Multiscale Relevancy Extraction.** To support Semantic Abstraction, we propose a multi-scale relevancy extractor for vision transformers [17] (ViTs), which robustly detects small, long-tail objects and achieves over $\times \mathbf{60}$ speed up from prior work [18].
- Two novel **Open-world 3D Scene Understanding** tasks (open-vocabulary semantic scene completion and visually obscured object localization), a data generation pipeline with AI2-THOR [1] simulator for the tasks, and a systematic open-world evaluation procedure. We believe these two tasks are important primitives for bridging existing robotics tasks to the open-world.

## 2   Related Works

**2D Visual Language Models.** The recent advancements in scaling up contrasting learning have enabled the training of large 2D vision language models (VLM). Through their exposure to millions of internet image-caption pairs, these VLMs [7, 9–11] acquire a remarkable level of zero-shot robustness – they can recognize a diverse set of long-tail semantic concepts robustly under distributional shifts. However, the learned image-level representations are not directly applicable to spatial-reasoning tasks which require pixel-level information. To address this issue, approaches were proposed to extract dense features [19–21] or relevancy maps [18, 22]. These advancements complement our contribution, which leverages such techniques for interfacing 2D VLMs to open-world 3D scene understanding tasks.

**Transferring 2D VLMs to 2D Applications.** Encouraged by their impressive cabilities, many downstream tasks build around these pretrained VLMs [14, 23–28], often finetuning them (either end-to-end or train learnable weights on top of the pretrained encoder) on a small task-specific dataset. However, many recent studies show that finetuning these VLMs significantly weakens their zero-shot robustness [7, 10, 12, 13, 19], which motivates a new paradigm for using pretrained models. By combining a *fixed* pretrained VLM with prompting [29] or attention extraction [15, 19–21], the new generation of zero-shot transfer techniques can inherit the VLM's robustness in their downstream tasks without overfitting to any visual domain. However, all existing zero-shot transfer techniques have been confined to the realms of 2D vision.

**Closed-World 3D Scene Understanding.** Understanding the semantics and 3D structure of an unstructured environment is a fundamental capability for robots. However, prior works in 3D object detection [30–32], semantic scene completion [33–35], and language-informed object localization [6, 36] are typically concerned with limited visual diversity and object categories, limitations imposed by their 3D training data. Despite great efforts from the community [1–6], existing 3D datasets' scale, diversity,

---

[1]All code, data, and models is publicly available at https://github.com/columbia-ai-robotics/semantic-abstraction.

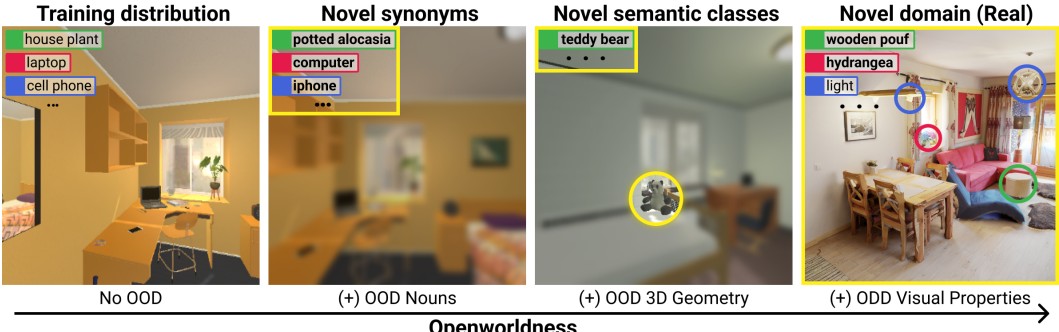

Figure 2: **Open-world generalization requirements** can build on top of each other, forming tiers of open-worldness (distinct property of each tier outlined yellow).

and coverage pale in comparison to internet-scale image-text pairs [7–9]. However, our key insight is that visual-semantic reasoning can be learned separately from 3D reasoning. That is, with the right abstraction, the complex visual-semantic reasoning can be offloaded to 2D VLMs, while the 3D model specializes in semantic-agnostic spatial and geometry reasoning.

## 3 Method: Semantic Abstraction

Evidence from linguistics and cognitive science suggests that semantic concepts are much more diverse than spatial and geometric concepts. While the average adult knows roughly 10,000 nouns for object categories, there are only 90 words in English for describing spatial relations [37]. In fact, there are so few spatial prepositions that they are typically considered a closed-class vocabulary [38]. Similarly for 3D geometry, the idea that our 3D world can be decomposed into sets of primitives [39] has been applied in many works, such as generalized cylinders [40] and block worlds [41]. In contrast, the space of nouns for semantic concepts is large and often cannot be further decomposed into common primitives. This suggests that while we may need a large and diverse dataset for semantic and visual concept learning, a relatively small 3D dataset could cover spatial and geometric reasoning. This observation motivates our approach, Semantic Abstraction.

### 3.1 Abstraction via Relevancy

The Semantic Abstraction (SemAbs) module (Fig 3c) takes as input an RGB-D image $\mathcal{I} \in \mathbb{R}^{H \times W}$ and an object class text label $\mathcal{T}$ (*e.g.* "biege armchair") and outputs the 3D occupancy $\mathcal{O}$ for objects of class $\mathcal{T}$. It factorizes into two submodules:

**The semantic-aware wrapper** (Fig 3c, green background) abstracts $\mathcal{I}$ and $\mathcal{T}$ into a relevancy map $\mathcal{R} \in \mathbb{R}^{H \times W}$, where each pixel's value denotes that pixel's contribution to the VLM's classification score for $\mathcal{T}$. Introduced for model explainability [18, 22], relevancy maps can be treated as a coarse localization of the text label. Using the depth image and camera matrices, $\mathcal{R}$ is projected into a 3D point cloud, $\mathcal{R}^{\mathrm{proj}} = \{r_i\}_{i=1}^{H \times W}$ where $r_i \in \mathbb{R}^4$ (a 3D location with a scalar relevancy value). Only $\mathcal{R}^{\mathrm{proj}}$, but neither the text $\mathcal{T}$ nor the image $\mathcal{I}$, is passed to the second submodule.

**The semantic-abstracted 3D module** (Fig 3c, grey background) treats the relevancy point cloud as the localization of a partially observed object and completes it into that object's 3D occupancy. To do this, we first scatter $\mathcal{R}^{\mathrm{proj}}$ into a 3D voxel grid $\mathcal{R}^{\mathrm{vox}} \in \mathbb{R}^{D \times 128 \times 128 \times 128}$. Then, we encode $\mathcal{R}^{\mathrm{vox}}$ as a 3D feature volume: $f_{\mathrm{encode}}(\mathcal{R}^{\mathrm{vox}}) \mapsto Z \in \mathbb{R}^{D \times 128 \times 128 \times 128}$, where $f_{\mathrm{encode}}$ is a 3D UNet [42] with 6 levels. $Z$ can be sampled using trilinear interpolation to produce local features $\phi_q^Z \in \mathbb{R}^D$ for any 3D query point $q \in \mathbb{R}^3$. Finally, decoding $\phi_q^Z$ with a learned MLP $f_{\mathrm{decode}}$ gives us an occupancy probability for each point $q$, $f_{\mathrm{decode}}(\phi_q^Z) \mapsto o(q) \in [0,1]$. In this submodule, only $f_{\mathrm{encode}}$ and $f_{\mathrm{decode}}$ (Fig. 3c, yellow boxes) are trained with the 3D dataset.

Although the semantic-abstracted 3D module only observes relevancy abstractions of the semantic label $\mathcal{T}$ but not $\mathcal{T}$ itself, the semantic-abstracted 3D module's output can be interpreted as the 3D occupancy for $\mathcal{T}$. This means it generalizes to any semantic label that can be recognized by the 2D VLMs' relevancy maps even if it was trained on a limited 3D dataset. In our implementation, we use CLIP [7] as our VLM. However, our framework is *VLM-agnostic*, as long as relevancy maps can be generated. It is a interesting future direction to investigate how different VLMs [7–11], their training procedure [43], and different relevancy approaches [18, 22, 44, 45] affect the performance of different downstream 3D scene understanding tasks. In the next sections, we explain how to ensure the visual-semantic concepts are reliably recognized by the relevancy maps (Sec. 3.2), and how to apply SemAbs to 3D scene understanding tasks (Sec. 3.3, Sec. 3.4).

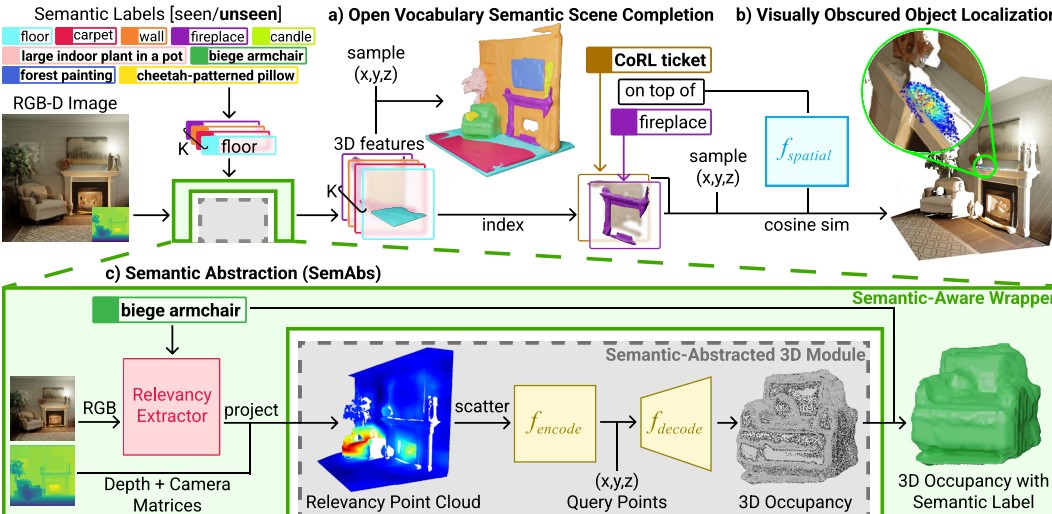

Figure 3: **Semantic Abstraction Overview.** Our framework can be applied to open-world 3D scene understanding tasks (a-b) using the SemAbs module (c). It consists of a **semantic-aware wrapper** (green background) that abstracts the input image and semantic label into a relevancy map, and a **semantic-abstracted 3D module** (grey background) that completes the projected relevancy map into a 3D occupancy. This abstraction allows our approach to generalize to long-tail semantic labels unseen (bolded) during 3D training, such as the "CoRL ticket on top of the fireplace".

## 3.2 A Multi-Scale Relevancy Extractor

We choose to use ViT-based CLIP [7] due to their superior performance over the ResNet variants. However, existing ViT relevancy techniques [18] often produce noisy, low-resolution maps that highlight irrelevant regions or miss small objects [27, 46].

To reduce noise from irrelevant regions, we use horizontal flips and RGB augmentations [27] of $\mathcal{I}$ and obtain a relevancy map for each augmentation. To reliably detect small objects, we propose a multi-scale relevancy extractor that *densely* computes relevancies at different scales and locations in a sliding window fashion (an analogous relevancy-based approach to Li et al.'s local attention-pooling). The final relevancy map is averaged across all augmentations and scales.

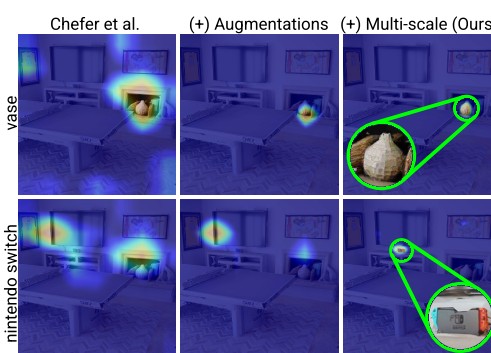

Figure 4: Our relevancy extractor robustly detects even small, long-tail objects, like the "nintendo switch".

Qualitatively, while augmentations help in reducing irrelevant regions (*e.g.* Fig. 4, from localizing the entire fireplace in the 1st column to only the vase in the 2nd column), they still often miss small objects. However, with our multi-scale relevancy extractor, our relevancy maps can detect small objects such as the tiny "nintendo switch" (Fig. 4, bottom right). To support efficient multi-scale relevancy extraction, we implemented a batch-parallelized ViT relevancy extractor. At $0.4$ seconds per text label on average[2], we achieve over $\times$ **60** speed up compared to the non-parallelized implementation. With the rise in popularity of ViTs, we believe this efficient ViT relevancy extractor implementation will be a useful primitive for the research community and have released it on Github and a Hugging Face Spaces.

## 3.3 Application to Open Vocabulary Semantic Scene Completion (OVSSC)

**Task.** Given as input a single RGB-D image $\mathcal{I}$ and a set of $K$ object class labels, represented as open-vocabulary text $\{\mathcal{T}_k\}_{k=1}^{K}$, the task is to complete the 3D geometry of all the partially observed objects referred by $\{\mathcal{T}_k\}_{k=1}^{K}$. In contrast to its closed-world variant [33], where the object list is fixed for all scenes, OVSSC formulation accounts for context-dependency, which allows the class labels to change from scene to scene and from training to testing. From this 3D semantic scene completion, robots can use the detailed 3D geometry for low level planning such as grasping and motion planning.

---

[2]measured over 10 repetitions with 100 text labels on a GTX 1080 Ti machine with a light load. We provide extra details in the appendix.

**Applying SemAbs to OVSSC** is a matter of applying the SemAbs module $K$ times, once per object label. First, the semantic-aware wrapper abstracts $\mathcal{I}$ and $\{\mathcal{T}_k\}_{k=1}^K$ into $K$ relevancy point clouds $\{\mathcal{R}_k^{\mathrm{proj}}\}_{k=1}^K$. Then, the semantic-abstracted 3D module extracts $K$ feature volumes $\{Z_k\}_{k=1}^K$, used to decode $K$ 3D occupancies $\{\mathcal{O}_k\}_{k=1}^K$. The final OVSSC output is a point-wise argmax of occupancy probability over the $K$ classes. Query points with the maximum occupancy probability less than a threshold $c$ are labeled as empty. We provide data generation and training details in the appendix.

### 3.4 Application to Visually Obscured Object Localization (VOOL)

**Motivation.** Even after completing partially visible objects in the scene, OVSSC's output may not be complete – the scene may contain visually obscured or hidden objects (e.g., the ticket on the fireplace in Fig. 3b). In such cases, the human user can provide additional hints to the robot using natural language to help localize the object, which motivates our VOOL task.

**Task.** Given as input a single RGB-D image $\mathcal{I}$ of a 3D scene and a description $\mathcal{D}$ of an object's location in the scene, the task is to predict the possible occupancies of the referred object. Similar to prior works [47], we assume that the description is in the standard linguistic representation of an object's place [37]. Specifically, $\mathcal{D} = \langle \mathcal{T}_{\mathrm{target}}, \mathcal{S}, \mathcal{T}_{\mathrm{ref}} \rangle$, where the open-vocabulary target object label $\mathcal{T}_{\mathrm{target}}$ (*e.g.* "rapid test") is the visually obscured object to be located, the open-vocabulary reference object label $\mathcal{T}_{\mathrm{ref}}$ (*e.g.* "harry potter book") is presumably a visible object, and the closed-vocabulary spatial preprosition $\mathcal{S}$ describes $\mathcal{T}_{\mathrm{target}}$'s location with respect to $\mathcal{T}_{\mathrm{ref}}$ (*e.g.* "behind"). Unlike the referred expression 3D object localization task [6] which assumes full visibility (*i.e.*, fused pointcloud), VOOL takes as input only the current RGB-D image, and thus is more suited for a dynamic, partially observable environment. Further, in contrast to bounding boxes, VOOL's occupancy can be disjoint regions in space, which accounts the multi-modal uncertainty when objects are hidden. Downstream robotic applications, such as object searching, benefit more from the uncertainty information encoded in a VOOL's output than bounding boxes.

**Applying SemAbs to VOOL.** We propose to learn an embedding $f_{\mathrm{spatial}}(\mathcal{S}) \in \mathbb{R}^{2D}$ for each spatial relation (Fig. 3b, blue box), since the set of spatial prepositions is small and finite [37], and use them as follows. As in OVSSC (§3.3), we can get occupancy feature volumes $Z_{\mathrm{target}}$ and $Z_{\mathrm{ref}}$ for the target and reference objects respectively. Given a set of fixed query points $Q = \{q_i\}_{i=1}^N$, we extract their local feature point clouds $\phi_Q^{Z_{\mathrm{target}}}, \phi_Q^{Z_{\mathrm{ref}}} \in R^{N \times D}$, then concatenate them to get a new feature pointcloud $\phi_Q^{Z_{\mathrm{target}} \| Z_{\mathrm{ref}}} \in R^{N \times 2D}$. Finally, to get the 3D localization occupancy $\mathcal{O}$, we perform a point-wise cosine-similarity between $\phi_Q^{Z_{\mathrm{target}} \| Z_{\mathrm{ref}}}$ and $f_{\mathrm{spatial}}(\mathcal{S})$. For this task, we only need to learn the spatial embeddings $f_{\mathrm{spatial}}$ conditioned on the occupancy information encoded in $Z_{\mathrm{target}}$ and $Z_{\mathrm{ref}}$, since semantic reasoning of $\mathcal{T}_{\mathrm{target}}$ and $\mathcal{T}_{\mathrm{ref}}$ has been offloaded to the SemAbs (and, therefore, to CLIP). We provide data generation and training details in the appendix.

## 4 Experiments

We design experiments to systemically investigate Semantic Abstraction's open-world generalization abilities to novel[3] concepts in 3D scene understanding. Our benchmark include the following categories: 1) **Novel Rooms**: we follow AI2-THOR [1]'s split, which holds out 20 rooms for as novel rooms, while training on 100 rooms. 2) **Novel Visual**: we use AI2-THOR's randomize material and lighting functionality to generate more varied 2D visual conditions of novel rooms. 3) **Novel Synonyms**: we manually picked out 17 classes which had natural synonyms (*e.g.* "pillow" to "cushion", complete list included in appendix) and replace their occurrences in text inputs (*i.e.*, object label in OVSSC, description in VOOL) to these synonyms. 4) **Novel Class**: we hold out 6 classes out of the 202 classes in AI2-THOR. For OVSSC, any view which contains one of these novel classes is held out for testing. For VOOL, any scene with a description that contains a novel class (as either the target or reference object) is held out. 5) **Novel Domain**: We also provide qualitative evaluations (Fig. 5) using real world scans from HM3D [4].

**Metrics & Baselines.** For both tasks, we measure the learner's performance with voxel IoU of dimension $32 \times 32 \times 32$, where the ground truth of each task is generated as described in §3.3 for OVSSC and §3.4 for VOOL. We compare with following categories of baselines:

- **Semantic-Aware (SemAware).** To evaluate the effectiveness of semantic abstraction, we design baselines that perform visual-semantic reasoning themselves by taking RGB point clouds as input

---

[3]Here, we define "novel" as to whether a concept was observed during the 3D training process.

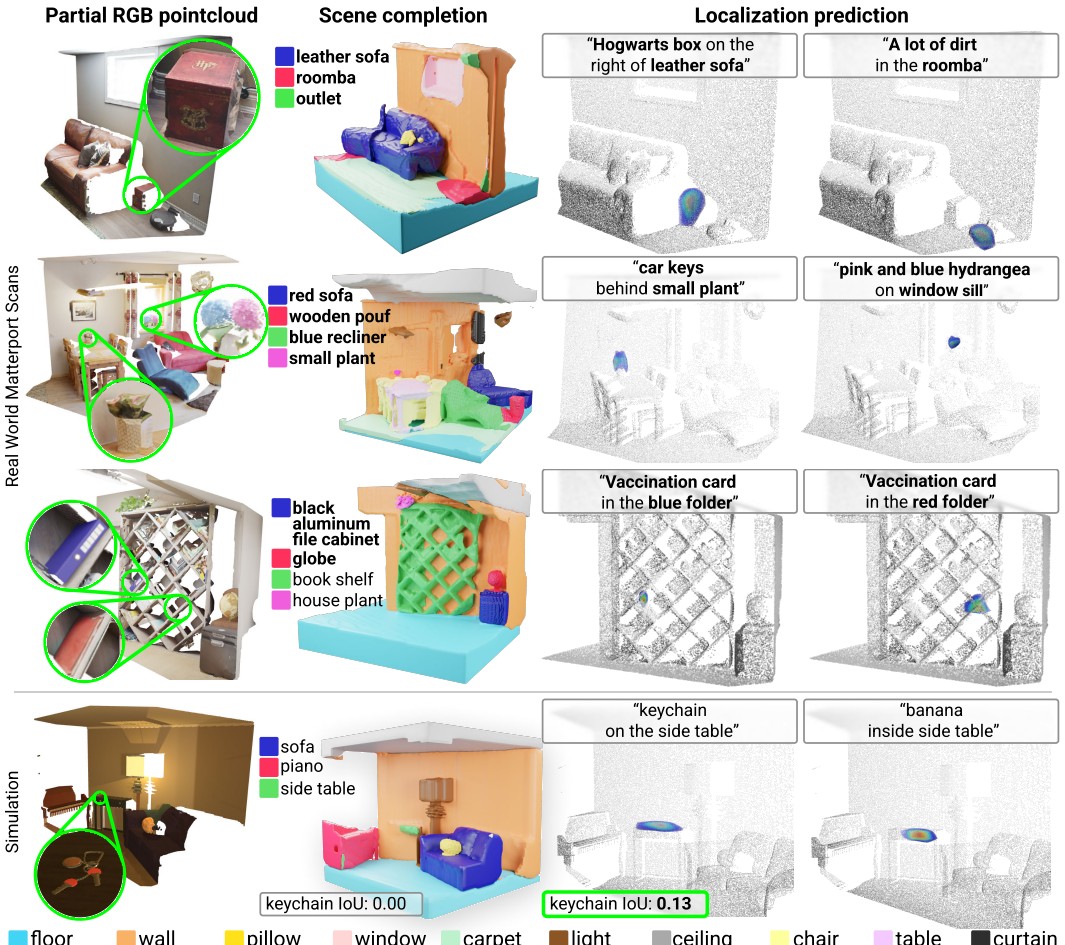

**Partial RGB pointcloud** | **Scene completion** | **Localization prediction**

Real World Matterport Scans

leather sofa
roomba
outlet

"Hogwarts box on the right of leather sofa" | "A lot of dirt in the roomba"

red sofa
wooden pouf
blue recliner
small plant

"car keys behind small plant" | "pink and blue hydrangea on window sill"

black aluminum file cabinet
globe
book shelf
house plant

"Vaccination card in the blue folder" | "Vaccination card in the red folder"

Simulation

sofa
piano
side table

"keychain on the side table" | "banana inside side table"

keychain IoU: 0.00 | keychain IoU: **0.13**

floor | wall | pillow | window | carpet | light | ceiling | chair | table | curtain

Figure 5: **Semantic Abstraction inherits CLIP's visual-semantic reasoning skils** From distinguishing colors (*e.g.* "red folder" v.s. "blue folder") to recognizing cultural (*e.g.* "hogwarts box") and long-tail semantic concepts (*e.g.* "roomba", "hydrangea"), our approach offloads such visual-semantic reasoning challenges to CLIP. Indoing so, its learned 3D spatial and geometric reasoning skills transfers sim2real in a zero-shot manner.

(instead of relevancy point clouds). For OVSSC, the baseline's feature point cloud is supervised with BCE to be aligned (*i.e.*, using cosine similarity) with the correct object class text embedding. Similarly, for VOOL, the feature point cloud's cosine similarity with an embedding of the entire localization description (*e.g.* "N95s inside the trash") is supervised using BCE to the target object occupancy.

- **Semantic & Spatial Abstraction (ClipSpatial).** What if we off-load both 3D and visual-semantic reasoning to 2D VLMs (instead of just the latter, as in our approach)? To answer this, we designed CLIPSpatial, a VOOL baseline that uses relevancy maps for the entire description (*e.g.* "N95s inside the trash") as input. We expect this approach to perform poorly since it has been demonstrated that current 2D VLMs [7, 8] struggles with spatial relations [14–16, 48].
- **Naive Relevancy Extraction. (SemAbs + [18])** To investigate the effects of relevancy quality, we replace our multi-scale relevancy extractor with Chefer et al. [18]'s approach.

The first two baselines can be seen as two extremes on a spectrum of how much we rely on pretrained 2D VLMs. While SemAware learns both semantics and 3D spatial reasoning, CLIPSpatial delegates both to pretrained 2D VLMs. Our approach is in between these two extremes, designed such that it builds on 2D VLMs' visual robustness and open-classification strengths while addressing its spatial reasoning weaknesses.

### 4.1 Open-world Evaluation Results

Tab. 1 and 2 summarize the quantitative results, and Fig. 5 shows real world qualitative results tested on Matterport scenes [4]. More results and comparisons can be found on the project website.

**Semantic Abstraction simplifies open-world generalization.** SemAware baselines especially struggle to generalize to novel semantic classes (Tab. 1, Tab. 2).

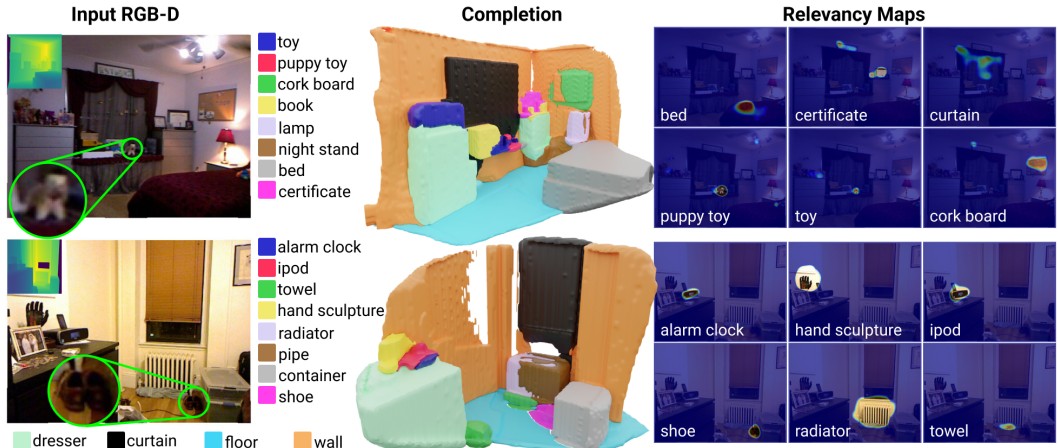

Figure 6: **Qualitative result on NYUCAD [49, 50] dataset.** Our model can distinguish between "puppy toy", "ipod", "certificate", and "hand sculpture", while all prior works would've output "obj".

We hypothesize that observing text embeddings during training causes these baselines to specialize to this distribution of embeddings, such that some synonyms and most novel class labels are out-of-distribution inputs. Our approach significantly outperforms the SemAware baseline across the board, suggesting that Semantic Abstraction not only simplifies open-world generalization but is also a strong inductive bias for learning the task.

| Approach | Novel | | | |
|---|---|---|---|---|
| | Room | Visual | Synonyms | Class |
| SemAware | 32.2 | 31.9 | 20.2 | 0.0 |
| SemAbs+[18] | 26.6 | 24.3 | 17.8 | 12.2 |
| Ours | **40.1** | **36.4** | **33.4** | **37.9** |

Table 1: **Open-Vocab Semantic Scene Completion**

**Semantic & Spatial Abstraction generalizes poorly.** The CLIPspatial baseline achieves 8.0-9.0 IoU worse than our approach in all generalization scenarios (Tab. 2). This indicates that our design of learning spatial reasoning instead of relying on CLIP was crucial to semantic abstraction's performance.

**Semantic Abstraction requires high quality relevancy maps.** Intuitively, the model can only output high quality completions if its relevancy maps are also high quality (*i.e.*, highlights the target object, especially when the object is small). Our average IoU is ×**1.5** (novel room, visual, synonyms, Table 1), over ×**2.0** (novel classes, Table 1, novel room, visual, synonyms, Table 2) or ×**5.0** (novel class, Table 2) compared to SemAbs + [18], which demonstrates the importance of our relevancy extractor in a successful application of Semantic Abstraction.

| Approach | Novel | | | |
|---|---|---|---|---|
| | Room | Visual | Synonyms | Class |
| SemAware | 12.1 | 11.9 | 11.4 | 3.0 |
| SemAbs+[18] | 9.1 | 8.8 | 10.7 | 4.0 |
| CLIPspatial | 11.7 | 10.1 | 14.3 | 11.2 |
| Ours | **20.9** | **19.2** | **23.4** | **19.7** |

Table 2: **Visually Obscured Object Localization**

**Inheriting VLM zero-shot robustness enables Sim2Real transfer.** Despite training in simulation, our approach can perform completion and localization from real-world matterport scans [4] (Fig. 1 and 5). From novel vocabulary for describing materials (*e.g.* "leather sofa", "wooden pouf"), colors (*e.g.* "red folder", "blue folder"), conjunctions of them (*e.g.* "black aluminum file cabinet") to novel semantic classes (*e.g.* "roomba", "Hogwarts box"), our approach unlocks CLIP's visual robustness and large repertoire of visual-semantic concepts to 3D scene understanding. When OVSSC's completion misses objects (*e.g.* "hydrangea", "keychain"), VOOL can still localize them (*e.g.* +0.13 IoU for keychain).

## 4.2 Zero-shot Evaluation Results

The NYUv2 CAD dataset [49, 50] is a popular closed-vocabulary SSC benchmark. Although it contains 894 object classes, prior works [33, 51, 52] collapse it down to 11 classes to avoid the challenge in learning the "long-tail" categories (*e.g.* ,"puppy toy","ipod", "hand sculture"). We evaluate our model zero-shot on NYUv2 directly on **all 894 classes** as well as the 11 classes for comparison (Fig. 6, Tab. 3, Ours). We also train a SemAbs model from scratch in NYU to investigate the generalization gap between our dataset and NYUv2 (Tab. 3, Ours NYU). We use the same training/testing split [33]. Similar to prior works [33], we report volumetric IoU at a voxel dimension of 60. As in our THOR dataset, we only train $f_{encode}$ and $f_{decode}$ in the semantic abstracted 3D module, not the 2D VLM. To our knowledge, our approach is the first to output the full list of 894 object categories in this benchmark.

| Approach | ceil. | floor | wall | win. | chair | bed | sofa | table | tvs | furn. | objs. | avg. |
|---|---|---|---|---|---|---|---|---|---|---|---|---|
| SSCNet [33] | 32.5 | 92.6 | 40.2 | 8.9 | 33.9 | 57 | 59.5 | 28.3 | 8.1 | 44.8 | 25.1 | 36.6 |
| SISNet [51] | 63.4 | 94.4 | 67.2 | 52.4 | 59.2 | 77.9 | 71.1 | 51.8 | 46.2 | 65.8 | 48.8 | 63.5 |
| Ours **Supervised** | 22.6 | 46.1 | 33.9 | 35.9 | 23.9 | 55.9 | 37.9 | 19.7 | 30.8 | 39.8 | 27.7 | 34.0 |
| Ours **Zeroshot** | 13.7 | 17.3 | 13.5 | 25.2 | 15.2 | 33.3 | 31.5 | 12.0 | 23.7 | 25.6 | 19.9 | 21.0 |

Table 3: **NYUv2 CAD Semantic Scene Completion.**

**Open-world versus Closed-world formulation.** Our work investigates open-world 3D scene understanding, which means specializing to one data distribution is not our focus. However, we hope these results on a standard closed-world benchmark along with our discussion below are helpful to the research community in understanding practical trade-offs in algorithm and evaluation design between the open-world versus closed-world formulation. Below summarizes our key findings:

+ **Flexibility to object labels.** Our framework directly outputs the full 894 object categories, of which 108 do not appear in the training split, and its predictions can map back to the collapsed 11 classes. This has never been attempted by prior works. This flexibility is achieved by building on top of 2D VLMs.

+ **Robust detection of small, long-tail object categories.** "Due to the heavy-tailed distribution of object instances in natural images, a large number of objects will occur too infrequently to build usable recognition models" [41]. However, our approach can robustly detect not just long-tail object classes, but also small ones – challenging scenarios for perception systems. For instance, from Fig. 6, our multi-scale relevancy extractor can output highly localized relevancy maps around "puppy toy","ipod", and "hand sculpture".

- **Inability to finetune large pretrained models.** As shown in prior works [7, 10, 12, 13, 19], directly finetuning 2D VLMs on small datasets hurts their robustness. We cannot finetune the visual encoders (and thus, its relevancy activations) to specialize on the visual and class distribution on a small dataset (e.g., NYUv2) without sacrificing it robustness and generality. This is an open and important research direction and recent work [53] has demonstrated promising results in image classification.

- **Bias from image caption data.** The typical training paradigm of current 2D VLMs [7, 10, 11] is contrastive learning over image-caption pairs on the internet. While their dataset covers a large number of semantic concepts, they still have their limits. For instance, we observed that for "wall" and "ceiling" – two classes which closed-world approaches typically achieve high performance for – our approach achieved an much lower IoU of only 33.9 and 22.6. Since these classes are often not the subject of internet captions, our approach inherit this bias from 2D VLMs. However, our framework is **VLM-agnostic**, which means it can continue to benefit from future and more powerful versions of 2D VLMs.

### 4.3 Limitations and assumptions

Our key assumption is that geometry and spatial reasoning can be done in a semantic agnostic manner. This assumption does not hold when we care about completing detailed object geometry, which is often semantic-dependent. However, we believe that coarse 3D reasoning with paired semantic labels may already useful for open-world robotic applications such as object retrieval. Second, while we have assumed a simple description syntax and a small set of spatial prepositions for object localization, incorporating natural language understanding (*i.e.*, to support open-set phrases for richer spatial descriptions) into open-world 3D scene understanding is an important extension of our work. Lastly, we assumed that relevancy maps highlight only relevant regions. Limited by current VLMs, our relevancy extractor can sometimes fail to distinguish highly related concepts (*e.g.* book v.s. bookshelf). However, since Semantic Abstraction is VLM-agnostic, this limitation could be addressed by using other VLMs.

## 5 Conclusion

We proposed Semantic Abstraction, a framework that equips 2D VLMs with 3D spatial capabilities for open-world 3D scene understanding tasks. By abstracting RGB and semantic label inputs as relevancy maps, our learned 3D spatial and geometric reasoning skills generalizes to novel vocabulary for object attributes and nouns, visual properties, semantic classes, and domains. Since scene completion and 3D object localization are fundamental primitives for many robotic tasks, we believe our approach can assist in bridging currently close-world robotics tasks, such as object retrieval, into the open-world formulation. More importantly, we hope our framework encourages roboticists to embrace building systems around large pretrained models, as a path towards open-world robotics.

**Acknowledgments:** We would like to thank Samir Yitzhak Gadre, Cheng Chi and Zhenjia Xu for their helpful feedback and fruitful discussions. This work was supported in part by NSF Award #2143601, #2132519 JP Morgan Faculty Research Award, and Google Research Award. The views and conclusions contained herein are those of the authors and should not be interpreted as necessarily representing the official policies, either expressed or implied, of the sponsors.

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
