# OpenReview forum: "Semantic Abstraction: Open-World 3D Scene Understanding from 2D Vision-Language Models"
_robot-learning.org/CoRL/2022/Conference — CoRL 2022 Poster_

### Official Review · Reviewer_ZWJc · 2022-07-01

**Originality:** Very Good
**Technical Quality:** Very Good
**Clarity Of Presentation:** Very Good
**Impact:** 4

**Recommendation:**

Weak Accept: I recommend accepting the paper, but will not argue for my recommendation if the majority of other reviewers have a different opinion.

**Summary:**

The paper proposes SemanticAbstraction; a framework that incorporates 3D spatial capabilities to 2D vision-language models while allowing said models to keep their zero-shot learning abilities. The framework is comprised of 2 modules:
- Semantic Aware Wrapper which abstracts the input image given a semantic label using ViT-CLIP into a relevancy map projected to 3D
- Semantic Abstracted 3D Module which converts the projected 3D relevancy map into 3D occupancy prediction using a 3D encoder-decoder based model operating on 3D voxel grids

The framework is trained on synthetic only data and evaluated on 2 main tasks:
- Open Vocabulary Semantic Scene Completion (OVSSC): predicting 3D occupancy of objects given a single RGB-D image with object labels
- Visually Obscured Object Localization (VOOL): predicting possible occupancy locations of referred objects given an input image
The proposed method is compared against baselines and performs favorably.

**Issues:**

- What does BCE in line 230 stand for?
- What's the motivation behind using a 3D CNN for encoding? Did the authors try something in the direction of SRT: https://srt-paper.github.io/ which would help side-stepping a number of problems relating to using voxel-based representations.

**Quality Of The Limitations Section:**

Limitations are addressed clearly

**Reviewer Expertise:**

4: The reviewer is confident but not absolutely certain that the evaluation is correct

**Robotics Focus:**

Relevant but unlikely to deploy to hardware in near future

**Strengths And Weaknesses:**

Strengths:
- The paper is very well written, easy to follow
- The work addresses an important problem for robotics; the ability to link language and visual models while being robust to long-tail distribution descriptions
- The experimental results are quite promising
- Releasing code, dataset and models is beneficial to the advancement of this area of research

Weaknesses & Questions:
- It is not very clear to the reviewer why the semantic abstracted 3D module is needed? Why not just use the 3D projected relevancy map as an output with some post-processing?
- Using 3D voxel grids as a basis for the occupancy prediction can be potentially problematic for accurately localizing objects of small scale. Furthermore, maintaining a reasonable voxel size with large environments can lead to large memory consumption. It would be nice to see some information regarding the memory required or ablations on larger environments with small objects perhaps
- Is the OVSSC task a realistic one? Normally one would either want the location of a single object (as in the VOOL task) for which a prerequisite is segmenting the environment (where one would not expect giving categories as input). My question here is what does this task help us understand that we could not before?
- The framework was only evaluated on self-constructed baselines. While I understand that this area is rather novel with not enough related work that one can directly compare against. On the other hand, there are similar datasets like ScanRefer that offer a benchmark for a similar problem (albeit bounding box regression instead of segmentation). Can one perhaps adapt one of the methods from there or compare the current approach on said dataset?


**Summary Of Recommendation:**

The paper covers a key topic in robotics and presents very nice results showing generalization capabilities from synthetic dataset to real-world.

I would like to thank the authors for their responses. Based on the discussion, I will maintain my score.

---

> ### Author Response · Authors · 2022-08-24
> **Response**
>
> > It is not very clear to the reviewer why the semantic abstracted 3D module is needed? Why not just use the 3D projected relevancy map as an output with some post-processing?
>
> It is non-trivial to design a image/ geometry processing algorithm which takes as input the relevancy maps for **a partially observed surface** and outputs an **complete** 3D occupancy representation, as our approach does.
>
>
> > What's the motivation behind using a 3D CNN for encoding? Using 3D voxel grids as a basis for the occupancy prediction can be potentially problematic for accurately localizing objects of small scale.
>
> Indeed, the scatter operation from 3D relevance point cloud to 3D volume R^{vox} results in some loss of high frequency spatial information. However, while high-frequency details are important for recognizing objects (i.e., for object detection), it is no longer crucial in the semantic abstracted 3D module.  At this stage, the algorithm has already partially localized the relevancy object, and the goal for the semantic abstracted 3D module is to complete the 3D geometries, which typically consists of low-frequency signals.
>
> Moreover, our approach outputs an implicit 3D occupancy, allowing us to upsample the output volume to higher resolution outputs.  Further, our results show that our method is able to localize small objects (e.g., rubik’s cube in Fig. 1, candle in Fig. 3, lemon on shelf in supplementary video).
>
> > Is the OVSSC task a realistic one? What does this task help us understand that we could not before?
>
> Thank you for the question! We do believe that OVSSC enables new capabilities that traditional perception algorithms lack.
>
> In the closed-world formulation, one typically decides on the list of relevant object categories before collecting/labeling data and training a model. In our open-world formulation, one can decide the list of relevant object categories after the fact without any retraining. In other words, our model can be deployed in different domains with a context dependent set of object categories in a zeroshot manner. This is one of CLIP’s biggest strengths, and our framework allows 3D scene understanding to inherit this strength.
>
> Within each domain, our approach could work off of a predefined list given to it at set up time. For instance, in home robot scenarios, the robot could be given a list of relevant objects in the home. Since CLIP relevancy activations are also indicative of when an object is not in view (see 1m57s in supplementary video), we expect that our approach would also work well in this setting.
>
> > The framework was only evaluated on self-constructed baselines. While I understand that this area is rather novel with not enough related work that one can directly compare against. On the other hand, there are similar datasets like ScanRefer that offer a benchmark for a similar problem (albeit bounding box regression instead of segmentation). Can one perhaps adapt one of the methods from there or compare the current approach on said dataset?
>
> Thank you for the question and the suggestion! Our formulation of the VOOL task allows for multi-modal outputs, which accounts for the ambiguous phrases (see “banana in cabinet” in supplementary video, 5m43s), and works from a single RGB-D image. This makes the task definition incompatible with ScanRefer’s bounding box and a full-observability assumption through a fused point cloud.
>
> However, we have provided extra experiment results on the standard NYUCAD benchmark for closed-world semantic scene completion  (Sec. 6.1, 6.4 Tab. 3, Fig. 6).  We refer to the revised manuscript for more details.
>
> > What does BCE in line 230 stand for?
>
> It stands for binary cross entropy. We have added this to the paper.

---

### Official Review · Reviewer_Vpqh · 2022-07-20

**Originality:** Good
**Technical Quality:** Good
**Clarity Of Presentation:** Fair
**Impact:** 2

**Recommendation:**

Weak Reject: I recommend rejecting the paper, but will not argue for my recommendation if the majority of other reviewers have a different opinion.

**Summary:**

This paper formulates and studies the task of open-world 3D scene understanding. The key insight of the authors is that the task of spatial/geometric reasoning can be decoupled from the task of learning semantic embeddings of images and text. The paper presents a method that decouples the semantic component (using pre-trained a CLIP transformer) from the spatial component (3D point cloud completion), and show results in a simulator as well as some real-world 3D scans, along with generalization results to new environments and textual classes.

**Issues:**

- [Points Raised Above] I raised some concerns in the above section and some clarification in response to those points would really help me re-evaluate the submission score.
	- Reproducibility concerns
	- Positioning wrt robotics
	- Experiments
- [Limited Spatial Understanding] The authors should consider toning down their “open-vocabulary” claims in light of the fact that (i) they expect a very specific type of phrase (Line 182) to describe the object in the scene, and (ii) they further reduce the set of spatial prepositions to just *six* (Line 195), which is much more restrictive. Generally speaking, the repeated claim of “small/finite words for describing spatial relations” [citing Landau and Jackendoff] is not entirely true and there are several other ways spatial relations may be described ([see commentary by Bennett on the same paper](https://ase.tufts.edu/cogstud/jackendoff/papers/what_where.pdf)). While this may not be a big limitation in practice, the current claims need to be qualified (or clarified) better to present a more accurate discussion of prior work and motivation.

**Quality Of The Limitations Section:**

Limitations are addressed clearly

**Reviewer Expertise:**

4: The reviewer is confident but not absolutely certain that the evaluation is correct

**Robotics Focus:**

Relevant but unlikely to deploy to hardware in near future

**Strengths And Weaknesses:**

Strengths
---
- The VLM relevancy module seems like a neat little idea by combining the GradCam-style visualization technique with vision transformers. To my best knowledge, this is the first work that demonstrates an effective way to do this, and I can think of several research areas that would benefit for this.
- Despite being restrictive, the way the authors handle the spatial prepositions seems very simple, yet unique. I have seen a lot of CLIP-based papers struggle with this and it's very exciting to see this work well enough for a downstream application.
- The evaluation benchmark, designed on the AI2-THOR simulator, seems very thorough and provides good insights on which components of the method contribute the most. This could be a valuable contribution to the 3D scene understanding community if made public along with reproducible baselines etc.
- The supplemental video does a very good job of conveying the results and qualitatively comparing to baselines — I really like it!

***
Weaknesses
---
- [Implementation Details & Reproducibility] My biggest concern with the paper in its current shape is the lack of implementation details in multiple sections throughout the paper. This, coupled with the lack of any code release makes the paper very hard to parse, reproduce, and draw any generalizable insights from (the authors mention “code, models will be publicly available” but my review focuses on the provided material). This further makes it difficult to judge what components of the method are novel and what components are used directly from prior (unreferenced) work. Below are some notable examples of crucial implementation/training details missing:
	- Section 3.3.1: There is no information here regarding the models or training pipeline. What is the learning objective, and how is it achieved? Is this a novel component, or do you borrow this from prior work?
	- Section 3.3.2: This is particularly hard to parse, since the “method” is barely described in text (how is the query point allocation/matching done? etc.), and there are absolutely no training details provided.
	- Section 3.2: There are some implementation details here but the important stuff is only mentioned in the supplemental material, making the text seem incomplete. Further, this is not referenced in the main text.
- [Robotics Focus] The paper reads a lot like a typical computer vision paper that creates and evaluates on a task-specific benchmark without considering effects on a potential downstream robotic task (in simulation or real). Barring the conclusion (Line 292), the paper makes no effort to justify why the method may be well-suited for an embodied navigation task and why the proposed benchmark and metrics matter — why is the voxel IoU a good metric for a semantic navigation task where the robot just needs to reach a general location and may not care about the exact 3D voxels? Especially to the CoRL audience, the paper would benefit from a better discussion of the motivation and positioning wrt applications in robotics, if not by adding a relevant downstream application.
- [Organization] In addition to missing implementation details, the paper is also missing any qualitative visualization of the comparative experiments and there is no qualitative or quantitative table/figure corresponding to the sim2real claim (Line 270) — this only makes an appearance in the supplemental material with only a placeholder in the main text. Maybe this is a late addition after the paper deadline, but in its current form, the paper just feels incomplete by itself and needs to be re-organized with more essential information in the main text.
- [Experiments/Baselines] While the authors do a good job of evaluating the method on the proposed suite of tasks, I would argue that the evaluation is not against any “baselines”, and are closer to a set of ablations to the proposed method (except the one with Chefer et al.). This is not immediately a bad thing, but the authors should consider an analysis with respect to existing algorithms that work with a stricter set of assumptions to get a sense for how well the method performs. For instance, one such comparison can be against a SoTA closed-vocabulary scene completion algorithm, to understand how well SemAbs (proposed) done in comparison and where it lacks — does the CLIP-based model perform just as well as a more carefully engineered system or is it a long way from the best 3D scene understanding algorithms that the community has been developing.


**Summary Of Recommendation:**

In my review, I raised a number of concerns regarding the current submission: primarily on the lack of implementation details and related reproducibility concerns and the inability to assess significance of claims for downstream robotics applications. While I really like some ideas proposed in the paper, in its current form, I cannot recommend acceptance of the paper, but I am very willing to engage with the authors in the discussion phase and reconsider my recommendation.

---

**Update**: The authors' responses have addressed my concerns regarding lack of implementation details and presentation, and the last round of revisions with qualified claims of "open-vocabulary" mostly addresses my concern as well. However, I maintain that the relevance of the findings (and the choice of metrics to determine superior performance) to eventual downstream applications is not very clear, especially for applications in robotics (per the CoRL guidelines). Contingent on the promised revisions to qualify the open-vocabulary claims and the added discussion of relevance to robotics, I do not have any strong issues with the paper. Erring on the side of being conservative, I am still keeping my original rating as I cannot (enthusiastically or weakly) recommend acceptance but would not argue against it.

---

> ### Author Response · Authors · 2022-08-24
> **Response (1/2)**
>
> We would like to thank the reviewers for their time and effort in helping us improve our work. We are glad that the reviewer appreciated our idea and results.
>
>
> [Organization]
>
> We emphasize that we have provided qualitative results to support our sim2real in the main paper Figure 1, 3, and 5, which are tested on Matterport dataset. We have added the reference to Figure 1 and 5 in (Sec.4.1 Sim2Real transfer paragraph) to clarify the connection to sim2real. Due to the page limit during submission, we include more qualitative results in supplementary material.
>
> [Robotic Focus]
>
> We believe open-world 3D scene understanding is a critical capability for robotics. As recognized by both reviewer ZWJc and jSHt, “It is one of the challenging tasks in robotics due to the limited size of 3D datasets but is crucial to real-world applications.”
>
> We have designed the tasks with robotic systems in mind. For example, we formulated VOOL to output *multi-modal distribution* when the object is hidden. This uncertainty information is more useful for object searching than a bounding box localization (for instance, from ScanRefer’s formulation). Similarly, we designed OVSSC to output detailed 3D occupancies, which is useful for lower-action planning. We have added these details to the problem formulation in the method section (i.e., section 3.3) of the revised pdf.
>
> [“Open vocabulary” claim in VOOL]
> Our formulation in the main paper assumes as input open-vocabulary target and reference object labels (L182-L183 of original pdf) and closed-vocabulary spatial preposition S. We have named this task “Visually-Obscured Object Localization”, **without claiming open-vocabulary/natural language understanding**. We agree that assuming a simple localization expression is one of the limitations of the work (discussed in L281-283 of original pdf), and extending it to natural language settings are important extensions of our work.
>
>
> [Experiments/Baselines]
>
> As per the reviewer’s suggestions for comparison with closed-world baselines, we have provided new experiments on the closed-world SSC task on NYUCAD (Sec. 6.1 in the revised manuscript). Thank you for the suggestion, we agree this experiment adds much value to the paper by providing the opportunity to discuss the pros and cons between open-world and close-world formulation.
>
> While SemAbs cannot outperform SoTA close-world algorithms (trained on this dataset for the exact 11 object categories), it provides a few critical and unique **advantages** that SoTA close-world algorithms do not have.
>
> - Flexibility on object labels. To accommodate close-world algorithms, the NYU dataset collapses its object label from 894 classes to 11. This benchmark design  could negatively impact SemAbs’ quantitative performance when the categories are collapsed into less meaningful terms e.g., objects, furniture.
> Instead, we first use the 894 class in NYU to perform scene completion, and then collapse them into 12 categories for evaluation.  To our knowledge, our work is the first method that attempted to output the full 894 class on the NYU dataset. We provide evaluation for both 12 classes (Tab 3) and  894 classes (Sec. 6.4).
>
>
> - Robustness on small objects. By building on top of CLIP’s zeroshot-robustness, our multi-scale relevancy extractor can detect even small objects of long-tail semantic classes (see “puppy toy”, “ipod”, and “shoe” in Fig. 6 of the revised pdf).
>
> The obvious **disadvantage** of SemAbs is that, as its current form, its 2D module (i.e., CLIP) cannot be finetuned to overfit a particular domain without losing its zeroshot-robustness, and therefore fail to overfit to the visual properties of some classes in NYU. Similarly, limited by the current VLMs (trained with image-caption pairs), these models cannot reliably localize room structure  such as “wall” and “ceiling”, which are not typically mentioned in captions on the internet.
>
> However, our main contribution – the framework of semantic abstraction – is VLM-agnostic, which means 3D scene understanding approaches which apply our framework can continue to benefit for free from future VLMs.

---

> ### Author Response · Authors · 2022-08-24
> **Response (2/2)**
>
>
>
>
> [Implementation Details & Reproducibility]
>
> 1. We have attached an annotated subset of our codebase, which contains the training script, network and dataset classes, and instructions for downloading our preprocessed NYUCAD dataset. The README.md walks through the codebase structure and links to where and how different approaches are implemented. We hope this is informative in understanding the approach.
>
> 2. We have revised the method section to include more implementation details, and answer some specific questions below:
> Model: Our f_encode is a 3D UNet with 5 levels. Our f_decode is a two layer MLP with hidden dimensions 64.
> OVSSC training details added to Sec. 3.3.1: In each batch, we sample `B` scenes and `K` classes within each scene. The input to our network is a batch of 3D volume in shape `B x K x 32x 32x32`. After the f_encode, the output of the network is 3D feature volume. From which we will sample M  query points to obtain their occupancy probabilities of shape `B x K x M x 1`.
>
> This is supervised with the ground truth occupancy label using binary cross entropy. The network weights are optimized using the AdamW optimizer with learning rate 5e-4. If a scene does not have at least `K` classes, we pad the relevant dimensions in the input/output tensors with zeros. In our experiments, we use a `B = 8`, `K = 16`, and `N, M = 15000`.
>
> We chose this pipeline because it is the de facto 3D backbone and its simplicity allows emphasis on the effect of relevancy abstractions.
>
> VOOL Training details added to Sec. 3.3.2: In each batch, we sample `B` scenes and `K` descriptions from each scene. Two relevance volumes are given to our network for the target and reference object respectively. We use the SemAbs module to extract feature volumes, which can be sampled from to give using query points of shape `B x K x M x 3` to give two `B x K x M x D` feature point clouds. After concatenating these point clouds along the feature dimension and passing it through f_decode, we perform point wise cosine similarity between our learned spatial relation embedding, which gives us an occupancy probability point cloud of shape `B x K x M x 1`.
>
> These occupancy points are supervised with the ground truth occupancy of the target object. Note that when the object is not visible, ground truth contains all target location that satisfied the description (Details on how to generate the ground truth for VOOL can be found in Appendix 7.4).
>
> Similar to our OVSSC training, if a scene does not have at least `K` descriptions, we zero-pad the relevant dimensions in the input/output tensors. We also use the same loss, optimizer, learning rate, `B`, `N`, `M` as in OVSSC, and `K = 8` for this task.
>
> How are query points chosen? For both tasks, we perform balanced sampling on the 3D scene such that there is roughly an even split between positive and negative labels for each object category. This means, for small objects like the “salt shaker” (see 5m42s in supplementary video) roughly half of the sampled points are inside or on the “salt shaker”. We’ve included more information on data generation in section 3.3.1 of the revised pdf.

---

> ### Comment · Reviewer_Vpqh · 2022-08-26
> **Reviewer Response to Authors**
>
> - _[Reproducibility, **OK**]_ I appreciate the additional information shared here and this addresses my concerns regarding this. I would strongly encourage the authors to present these (in whatever format they deem fit) in a supplemental material or code README, along with a discussion of important/sensitive hyperparameters etc.
> - _[Positioning wrt Robotics, **Pending**]_ I appreciate the response, but this does not answer some of the questions I raised in my initial review, e.g. why do the metrics matter for a robotics application (or what a downstream use case may look like), why is voxel IoU an effective measure of performance etc. To emphasize the concern, the [CoRL guidelines](https://corl2022.org/instructions-for-authors/) very clearly state "All CoRL submissions must demonstrate the relevance to Robot Learning through Intent ... or Outcome", and "Submissions should focus on a core robotics problem and demonstrate the relevance of proposed models, algorithms, data sets, and benchmarks to robotics". This is not very clear to me.
> - _[Evaluation, **OK**]_ This was a great discussion and I would like to see it reflected in a revised PDF (in whatever capacity the author deem fit). The unique advantages and disadvantages of using a module like this deserve to be discussed in the main text, since they are quite literally the main takeaways of reading a new paper like this. I have no further concerns in this regard.
> - _[Limited Spatial Understanding, **Pending**]_ The "open-vocabulary" claim in the current presentation goes beyond just naming the modules. From early on (Abstract, Introduction), the authors frequently over-claim the capabilities in terms of supporting open-set vocabulary inputs for scene understanding. I understand what the authors are doing, and its doing a great job, but please consider qualifying these claims to what they are -- the "open vocabulary"-ness of the method is limited to associating objects (and not arbitrary phrases) to text. The text and figures somewhat mislead the reader to suggest that they can accept "open-set" phrases, whereas the method accepts a very specific kind of structure in the phrase, with (agreeably open vocabulary) objects and (limited vocabulary) information about spatial relations.

---

> > ### Author Response · Authors · 2022-08-27
> > **Revised PDF, Robot Object Search Application, and Clarified Open-vocabulary Scope**
> >
> > We have attached the second revision to our PDF with all changes highlighted in red. Please see below for our responses for each concern.
> >
> > > [Positioning wrt Robotics]
> >
> > **Downstream Applications**. We contextualize our two tasks in the application of object search from language guidance. For instance, using the object search example from [Kurenkov et al, ICRA 2021](https://ai.stanford.edu/mech-search/hms/), when asked to fetch `the jasmine green tea in the cabinet`, the robot would first use VOOL to compute the multimodal distribution of all possible locations of the tea (i.e. inside all the cabinets). This multi-modal uncertainty can inform the robot to selectively (e.g. starting with most likely region first) choose cabinet doors to open to reduce its uncertainty. Throughout this entire process, the robot can use OVSSC to simultaneously output 1) *accurate 3D geometry* required for low level planning such as grasping (e.g., [Varley et al IROS 2017](https://arxiv.org/abs/1609.08546)) or path planning (e.g., [Schmid et al 2022](https://arxiv.org/pdf/2208.08307v1.pdf)) and 2) *semantic labels* for objects which are required for higher level planning of which object to grasp. Specifically, even after opening the right cabinet, the robot would need to tell apart the `jasmine green tea` box from other boxes and cans in the cabinet (see Fig.1 in [Kurenkov et al, ICRA 2021](https://ai.stanford.edu/mech-search/hms/)),  *even* if it has never seen a `jasmine green tea` box in its 3D training dataset before.
> >
> > **Metric**. From this example of a downstream application, we believe a key requirement for the metric for the VOOL task is that it can evaluate multimodal predictions and outputs. Therefore, we choose voxel IOU for its intuitiveness and ubiquity in the field. However, we do agree that voxel IOU is not the only or optimal metric. On the other hand, Voxel IoU is a standard metric ever since the first semantic scene completion work [[Song et al, CVPR 2016](https://arxiv.org/pdf/1611.08974v1.pdf)], and we inherit this metric for OVSSC from its closed-world counterpart.
> >
> > > [Limited Spatial Understanding]
> >
> > **Novel Object Attributes and Object Nouns**. Although we use “open-vocabulary” to refer to a similar level of generalization requirement as prior works, (i.e. to novel object nouns, such as in [Gu et al ICLR 2022](https://openreview.net/forum?id=lL3lnMbR4WU)), we have added clarifications that in our work that we consider open-vocabulary to mean novel object attributes and object nouns. While the boundary between novel object attributes and novel phrases can be blurred with complex object attributes, such as “wall with colorful cartoon murals” (Fig. 1), “large indoor plant in a pot” (Fig. 2), “hair dryer with its wires tangled” and “upholstered chair in faux leather” (supplementary video, 3m36s), we will err on the conservative side and consider these to be novel object attributes.
> >
> > **Revision**. We’ve attached a second revision of our paper (all changes highlighted in read) to rehash our scope of open-vocabulary (`L11-12`, `L22`, `L332-333`), also changed our “Novel Vocab” quantitative evaluations to “Novel Synonyms” (`L253`, `Fig. 2`, `Table 1, 2, 4`). Further, we have added clarifications of our original limitation “incorporating natural language understanding” to mean “to support open-set phrases for richer spatial descriptions” in `L323-324`. Thank you for bringing up this potential point of confusion, and please let us know if our scope of open-vocabulary-ness is sufficiently clear from the revision.
> >
> > > [Evaluation]
> >
> > We also think the discussion of open-world v.s. closed-world formulation should deserve a prime location in the final paper, and we will restructure the paper to reflect this after the rebuttal process.

---

> > > ### Comment · Reviewer_Vpqh · 2022-08-28
> > > **Thanks, Final Comment**
> > >
> > > Thanks for the very quick response and comments. This has been a very productive discussion and mostly resolves my concerns. I will take this into advisement during the next phase of AC/reviewer discussions.

---

### Official Review · Reviewer_jSHt · 2022-07-20

**Originality:** Very Good
**Technical Quality:** Very Good
**Clarity Of Presentation:** Excellent
**Impact:** 3

**Recommendation:**

Weak Accept: I recommend accepting the paper, but will not argue for my recommendation if the majority of other reviewers have a different opinion.

**Summary:**

By combining the robustness in open-vocabulary scene understanding from a vision-language model in RGB images and a semantic-agnostic spatial understanding from a 3D model, it achieves open-world 3D scene understanding with a small set of 3D datasets.
It proposes two tasks completing partially observed objects and localizing hidden objects to show out-of-distribution generalization and spatial understanding ability on the partially observed 3D dataset. It provides various ablation models to help understand the effect of each part of the model.

**Issues:**

In general, evaluation is a little bit confusing; the first task, open vocabulary semantic scene completion (OVSSC), seems to be similar in open vocabulary detection/segmentation and the second task, visually obscured object localization (VOOL), can be considered as a specialized case of visual grounding. Besides the ablations, it lacks direct comparison to existing models or tasks. For instance, applying the proposed approach to Refer-it-in-RGBD (https://arxiv.org/abs/2103.07894) might be interesting to see how different approaches to spatial understanding work. How the model works in the existing detection tasks is another task of interest.

* Sec 3.3; how the 3D semantic-abstracted module is robustly working with ambiguous saliency or relevancy from the previous module.
* Hard to understand the effect of the Z_target in the VOOL task;
* How are query points Q chosen?
* Why 32x32x32? 3D detection methods can use refined volume up to 1 cm resolution.
* Why AI2-THOR? AI2-THOR is less diverse in terms of appearance, object classes and instances in each class, and types of rooms. While arguing its novelty in out-of-distribution generalization in vocabulary and appearance, testing only on AI2-THOR makes this approach less compelling.
* Visual grounding methods in either
a referring expression comprehension task or a referring expression segmentation task should be compared as a baseline such as `LAVT: Language-Aware Vision Transformer for Referring Image Segmentation', `CRIS: CLIP-Driven Referring Image Segmentation', `OFA: Unifying Architectures, Tasks, and Modalities Through a Simple Sequence-to-Sequence Learning Framework' (https://paperswithcode.com/sota/referring-expression-segmentation-on-refcoco-4, https://paperswithcode.com/sota/referring-expression-comprehension-on-refcoco) in order to show the evidence of the statement argued in line 237. Some of the spatial relationships can be interpreted in 2D (such as ``left of'') and it would be interesting to compare two different approaches that have different levels of task decompositions. At least, some 2D visual grounding models can be used in ClipSpatial.
* Instead of taking a single expression per instance, is choosing K objects at once in the OVSSC task affect the result? I wonder if exclusion happens or is encouraged during the process.
* Known-class accuracy (in comparison to existing detection methods) is missing.
* While being trained on six spatial prepositions, no information on spatial prepositions in the custom dataset is not provided; providing a success rate on each preposition may help better understand the problem.

* Typo in line 221: FOr > For

**Quality Of The Limitations Section:**

Limitations are addressed clearly

**Reviewer Expertise:**

3: The reviewer is fairly confident that the evaluation is correct

**Robotics Focus:**

Highly relevant to robotics but no hardware experiments

**Strengths And Weaknesses:**

Strengths
* Provides a simple but powerful decomposition and abstraction of perception and localization for out-of-distribution detection in robotics applications.
* Can leverage various VLMs as its module.
* Uses multi-scale proposals from VLMs to handle small objects with batched high-speed inference

Weaknesses
* Ambiguous evaluation on newly proposed datasets and tasks; not clear why they are proposed other than the existing tasks such as detection or visual grounding.
* Does not compare with existing methods in detection or visual grounding tasks; hard to understand the empirical contribution to real-world applications.
* Proposed datasets on AI2-THOR which is less diverse in appearance than other datasets such as ScanNet or SUN-RGBD.

**Summary Of Recommendation:**

It proposes a modular approach for 3D understanding with out-of-distribution generalization by combining vision-language model (VLM) with a semantic-abstract 3D model.
It is one of the challenging tasks in robotics due to the limited size of 3D datasets but is crucial to real-world applications.
It shows that the proposed method provides better generalization both in detecting small objects by introducing multi-scale feature extraction from VLM and identifying novel and fine-grained classes in 3D.
While its application is crucial for advancing the 3D understanding, its evaluation is not based on the existing tasks such as 3D detection or visual grounding, nor on existing approaches; therefore, it is not trivial to clearly understand their contribution compared to the existing literature.
Therefore, I recommend a weak acceptance.

---

> ### Author Response · Authors · 2022-08-24
> **Response (1/2)**
>
> Thank you for the detailed feedback, we are happy that the reviewer recognized our method as “simple but powerful”, and that it addresses a “challenging task in robotics.” The response below provides our response, clarifications, and additional experiments.
>
> [Evaluation]
>
> >Testing only on AI2-THOR makes this approach less compelling.
>
> We emphasize that we provide evaluation on **three** different datasets (in the original submission) and **four** in the updated manuscript, including ARKitScenes, Matterport-HM3D, and NYUCAD, which are real world scans/RGB-D images. These datasets differ from each other significantly along all axes we consider in open world generalization.
>
> > Why AI2-THOR?
>
> Towards a *systematic* procedure for open-world generalization evaluation, we needed a dataset which allowed us to separately control each dimension for open-worldness. Since prior real world datasets such as ScanRefer do not offer us this level of control, we opted for a simulator which provided us with a sufficient number of object classes and houses.
>
> > Visual grounding methods & Use 2D visual grounding models in ClipSpatial.
>
> Thank you for the suggestion and reference! Our formulation of the VOOL task outputs a 3D occupancy rather than a 3D bounding box. This generic problem formulation allows for multi-modal outputs in cases of ambiguous localization expressions. This formulation makes our task and prior localization/visual grounding/ referring expression tasks incompatible.
>
> However, we can indeed use LAVT in place of CLIPSpatial as suggested by the reviewer. We have included both quantitative (Table 2) and qualitative results (Fig. 7) in the revised pdf. Namely, we evaluated a baseline which is identical to ClipSpatial but uses LAVT’s output instead of CLIP’s relevancy as input, which we named LAVTSpatial. From Figure 6, LAVT struggles to robustly localize objects, on top of having no 3D spatial understanding. This is reflected in its poor quantitative performance in Table 2.
>
> This result indicates that despite having superior 2D spatial understanding on RefCoCo+ compared to CLIP, LAVT can’t generalize to the wide range of visual scenarios CLIP can, and thus serves as a poor abstraction mechanism for semantic abstraction. Please see Sec. 6.2 in the revised pdf for more details. Thank you for the suggestion, we agree this experiment added much value to our work.
>
> > Known-class accuracy.
>
> We have reported results on known-class performance in the first column (i.e, Novel Rooms) in both tables 1 and 2, which have the same set of object categories as training object categories.
>
> As discussed above, it’s not clear how to compare with existing detection methods due to different task definitions, however we do provide additional experiments with existing closed-world semantic scene completion methods on the standard NYUv2 dataset (Sec. 6.1). While our method does not outperform SOTA “closed-world” methods (which are allowed to overfit on one visual distribution for only 11 object categories), we want to emphasize its generality – it can be applied to any RGB-D dataset for any object categories without finetuning, a capability that no existing method possess. We have included a detailed discussion for this result in section 6.1 of the revised manuscript.
>
>
> [Method Clarification]
>
> > Sec 3.3; how the 3D semantic-abstracted module is robustly working with ambiguous saliency or relevancy from the previous module.
>
> Indeed, the relevancy map could contain ambiguities and uncertainties. However, we hypothesize that our 3D network learns to accommodate this distribution of VLM relevancy activations during training. This means at test time, if the VLM’s relevancy is in the same distribution as at training time, our approach should generalize, regardless of the underlying RGB image / semantic labels.
>
> > Hard to understand the effect of the Z_target in the VOOL task
>
> The VOOL include tasks where the target objects “hidden”, “visible” and anything in between. Therefore, Z_target allows our SemAbs module to convey its confidence of whether and where an object is in the scene. For instance, when the target object is clearly visible, its relevancy activations are strong and highly localized, and Z_target will provide the best estimate of the object’s occupancy. In another case, if the object is not visible, there may not be any relevancy activations, and Z_target will reflect this information.

---

> > ### Comment · Reviewer_jSHt · 2022-08-28
> > **Reviewer Response to Authors**
> >
> > First, I appreciate the responses with an additional experiment that mostly resolved my questions. Still, I have some questions or comments.
> >
> > ### VLM results seem to be too bad and not the best result from a pure VLM.
> > Though LAVT showed poor results on the grounding task as shown in Figure 7 in the revised submission, I still doubt that this truly shows the ability of VLM in zero-shot 2D visual grounding.
> > I have tried the example image from the revised submission with two sentences "plant on top of the bookshelf" and "light switch to the left of the door" with the OFA demo (https://huggingface.co/spaces/OFA-Sys/OFA-Visual_Grounding), I was able to produce these two results: https://imgur.com/a/j5RPOKv, https://imgur.com/a/pPGMHd4.
> > Even though OFA is designed for different tasks, it is hard to admit that the low scores from LAVT prove the better generalization power of the proposed method.
> > The point is, as the proposed method is using zero-shot VLM, to show the effect of the SemAbs in 3D scene-completion/localization in comparison to a naive VLM approach with depth.
> > And it is hard to confirm the performance contribution of the proposed approach in the object localization task.
> >
> > ### Multiple datasets
> > I could not find proper evaluation on multiple datasets other than AI2THOR in Table 1 and Table 2 and NYU semantic scene completion dataset which is additionally provided in the revised submission. I can understand that the training from AI2THOR was inevitable but still hard to confirm the argument on the sim2real transfer. I believe it can be generalizable to some extent, but still not sure if it will achieve a comparable number in zero-shot transfer unless the experiment is actually done.
> >
> > ### Additional questions
> > * What is `nan` in line 422, Section 6.4?
> > * Where can I find the ARKitScene dataset in the paper?
> > * The average number of SSCNet seems to be incorrect: 40.0 in Table 2, https://arxiv.org/pdf/2104.03640.pdf. (With a rough averaging even 40.0 might be incorrect.)

---

> ### Author Response · Authors · 2022-08-24
> **Response (2/2)**
>
>
> > How are query points Q chosen?
>
> During training, we perform balanced sampling on the 3D scene such that there is an even split between positive and negative labels for each object category. This means, for small objects like the “salt shaker” (see 5m42s in supplementary video) roughly half of the sampled points are inside or on the “salt shaker”. To ensure there are enough positive points on small and/or thin (e.g. magazine) objects, we also sample points on object’s mesh surfaces. We’ve included more information on data generation in section 3.3.1 of the revised pdf.
>
> > Why 32x32x32?
>
> Indeed, it is a lower input resolution than most object detection methods. However, we want to point out  that this volume is the input to the “semantic abstracted” 3D module, **after the object is localized** in the relevance map.  While high-frequency details in high-res input are important for recognizing objects, it is no longer crucial for the 3D module, whose task is completing geometries, which mostly consists of low-frequency signals.
>
> > Does choosing K objects at once in the OVSSC task affect the result?
>
> Exclusion (i.e., where different classes overlap spatially but only one of them is chosen) does happen, but to a similar degree as prior works (such as CCPNet, the 2019 SOTA for SSC on the NYUv2 benchmark) when they apply an argmax operator over their softmaxed one-over-K decoding head.
>
> >  Providing a success rate on each preposition
>
> We have added Tab. 4, and discussed the performance breakdown in Sec. 6.3.

---

### Author Response · Authors · 2022-08-24
**Response summary, revised PDF, and code .zip file**

**Comment:**

We would like to thank the area chair and the reviewers again for their time and insightful feedback to help us improve our paper. We are glad the reviewers thought our approach was a “simple but powerful” and unique way of “tackling challenging yet crucial tasks in real world robotics”, and that there will be “several research areas which could benefit from our work.”

1. We have added two experimental results for quantitative comparisons:
    - Evaluation on the closed-world SSC task on standard NYUCAD benchmark  (Sec. 6.1, 6.4 Tab. 3, Fig. 6).
    - Comparing a SOTA visual grounding model in our VOOL task (Sec. 6.2 and updated Tab. 2, LAVTSpatial row) and qualitative comparisons Fig. 7.
 2. We have extended the method section to include extra implementation and training details. All additions to the paper are in blue.
 3. We have attached a zip file of the training and evaluation code, along with instructions to download our preprocessed NYUCAD dataset and model weights.

We hope our revised writing, code release, and experiment additions helps in clarifying our approach and enabling comparison to prior works. We have addressed the reviewer’s questions and concerns (including the expression “open-vocabulary” concern) below.

For convenience, [the **original supplementary video** can be accessed at this link](https://drive.google.com/file/d/1VdKv0eQPykfp2iOXyiE0TE3P1IT1Bhzo/view?usp=sharing).


**Zip File:**

/attachment/32c42d129fddf89a3392230a64e12056fa4723cf.zip

---

### Author Response · Authors · 2022-08-28
**Reponse to reviewer jSHt's latest comments**

We thank reviewer jSHt’s latest comments and clarifying questions (posted 11 hours ago). Since the paper comments are now locked (and hidden from us), we will respond to their questions here in the main paper.

### SOTA VLM Comparison

We chose LAVT because it was explicitly suggested by reviewer jSHt in their original review (“Visual grounding methods in either a referring expression comprehension task or a referring expression segmentation task should be compared as a baseline such as `LAVT: Language-Aware Vision Transformer for Referring Image Segmentation`") and is [the current state of the art of referring expression segmentation on RefCOCO+](https://paperswithcode.com/sota/referring-expression-segmentation-on-refcoco-4).

> The proposed method is using zero-shot VLM to show the effects of the SemAbs in 3D scene-completion/localization in comparison to a naive VLM approach with depth.

We would like to reemphasize that LAVT is not `a naive VLM approach` (but instead a SOTA visual grounding model), and the `LAVTSpatial` baseline reported in Table 2 of the revised PDF is not depth-projected LAVT masks, but instead given the same 3D module built on top of a 3D U-Net architecture, identical to that of the SemAbs module and CLIPSpatial.

### Multiple datasets

We provide qualitative evaluations (L223 and L246, original PDF) on Matterport and ARKitScenes, because they do not provide ground truth 3D SSC labels.

Our SemAbs module is trained with the assumption that the RGB, depths, and ground truth 3D geometries are aligned. However, this assumption is broken in the NYU SSC benchmark, since the SSC ground truth are actually **approximations** produced by [Guo et al](https://arxiv.org/abs/1504.02437). This means, for the NYUv2 SSC benchmark (using kinect depth), the RGB and depths **are not** aligned with the 3D ground truth label (as pointed out by [Song et al](https://arxiv.org/pdf/1611.08974v1.pdf), section 5). On the other hand, for the NYUCAD SSC benchmark (using synthetic rendered depths), the RGB aren’t aligned with the depth and 3D geometry. Since they are computed from RGB, the relevancy map’s misalignment is lower bounded by that between the RGB and the depths/3D geometry.

One example of this is the `key` class.
> What is nan in line 422, Section 6.4

We get NaNs (not-a-number) in computing IoUs when the unions is zero, leading to a division by zero. The reason the IoU for `key` is `NaN` is because **all** scenes where keys appear in the visual input, there are **no** approximated ground truth label in the NYU SSC benchmark which corresponds to those keys.

We believe taking a model trained on **aligned** RGB, depths, and 3D geometries and directly evaluating it on **misaligned** depths and 3D geometries is not a useful comparison. Short of real world SSC benchmarks with aligned ground truth 3D labels, we have provided qualitative evaluations and comparisons on both Matterport and ARKitScenes.

###  Where can I find the ARKitScene dataset in the paper?

The download instructions for the dataset is  on [the dataset’s repo on Apple’s Github](https://github.com/apple/ARKitScenes).

### Incorrect SSCNet numbers

Indeed, Cai et al (the authors of [SISNet](https://arxiv.org/pdf/2104.03640.pdf)), Zhang et al (the authors of [CCPNet](https://arxiv.org/pdf/1908.00382.pdf)), Li et al (the authors of [AICNet](https://arxiv.org/pdf/2004.02122.pdf)), and Li et al (the authors of [DDRNet](https://arxiv.org/pdf/1903.00620.pdf)) have reported the wrong numbers for `wall`. At some point, a prior work made a typo from `49.2` to `40.2` for the wall category. We will update the correct `49.2` number (from [Garbade et al](https://arxiv.org/pdf/1804.03550.pdf)) after the rebuttal process, which gives the correct average IoU.

---

### Meta-Review · Area_Chair_du2e · 2022-08-14

**Recommendation:** Accept (Poster)
**Confidence:** 5

**Metareview:**

All reviewers agree that the paper addresses important issues in robot learning.
1. Although the method is simple, the experimental results are promising.
2. The authors constructed a new dataset for this task.
3. The paper and supplemental appendix are well organized. The supplemental video does a good job of conveying the qualitative results.

However, the reviewers raised a number of concerns.
1. The experimental comparison was conducted using only self-constructed baselines. It would have been helpful if the paper also presented evaluation on similar datasets.
2. Implementation details are not fully presented in the paper.
3. The expression "open vocabulary" should be reconsidered because spatial prepositions are limited to six.

### Post-rebuttal comment
The reviewers initially raised a few concerns regarding the experimental comparison and the implementation details. They also questioned some of the claims made in the initial submission. The authors provided a detailed response to each of these points and made changes to the paper to resolve most of the concerns. Therefore, I recommend acceptance of the paper.

**Best Paper Nomination:**

No